# Dysregulation of the hippocampal neuronal network by LGI1 auto-antibodies

Elodie Fels[1,2,3], Marie-Eve Mayeur[1,2,3], Estelle Wayere[1,2,3], Clémentine Vincent[1,2,3,4], Céline Malleval[1,2,3], Jérôme Honnorat[1,2,3,4], Olivier Pascual[1,2,3]*

1 Synaptopathies and Auto-antibodies (SynatAc) Team, Institut NeuroMyoGène, MeLis—CNRS UMR 5284—INSERM U1314, Universités de Lyon, Université Claude Bernard Lyon 1, Lyon, France, 2 Université Claude Bernard Lyon 1, Universités de Lyon, Lyon, France, 3 MeLis—CNRS UMR 5284—INSERM U1314, Lyon, France, 4 French Reference Center on Paraneoplastic Neurological Syndromes and Autoimmune Encephalitis, Hospices Civils de Lyon, Hôpital Neurologique, Bron Cedex, France

* olivier.pascual@inserm.fr

**Data Availability Statement:** All relevant data are within the article and its supporting information files.

**Funding:** This work was supported by FRM (Fondation pour la recherche médicale)

## Abstract

LGI1 is a neuronal secreted protein highly expressed in the hippocampus. Epileptic seizures and LGI1 hypo-functions have been found in both ADLTE, a genetic epileptogenic syndrome and LGI1 limbic encephalitis (LE), an autoimmune disease. Studies, based mainly on transgenic mouse models, investigated the function of LGI1 in the CNS and strangely showed that LGI1 loss of function, led to a decreased AMPA-receptors (AMPA-R) expression. Our project intends at better understanding how an altered function of LGI1 leads to epileptic seizures. To reach our goal, we infused mice with LGI1 IgG purified from the serum of patients diagnosed with LGI1 LE. Super resolution imaging revealed that LGI1 IgG reduced AMPA-R expression at the surface of inhibitory and excitatory neurons only in the dentate gyrus of the hippocampus. Complementary electrophysiological approaches indicated that despite reduced AMPA-R expression, LGI1 IgG increased the global hyperexcitability in the hippocampal neuronal network. Decreased AMPA-R expression at inhibitory neurons and the lack of LGI1 IgG effect in presence of GABA antagonist on excitability, led us to conclude that LGI1 function might be essential for the proper functioning of the overall network and orchestrate the imbalance between inhibition and excitation. Our work suggests that LGI1 IgG reduced the inhibitory network activity more significantly than the excitatory network shedding lights on the essential role of the inhibitory network to trigger epileptic seizures in patients with LGI1 LE.

## 1. Introduction

Leucin-rich glioma inactivated protein 1 (LGI1) is a 60 KDa secreted protein largely expressed in the central nervous system (CNS) with a high expression profile in the hippocampus. *LGI1* gene mutations are involved in an inherited form of epilepsy called autosomal dominant temporal lobe epilepsy (ADLTE) [1, 2]. Epileptic seizures are characterized by hyperexcitability and hypersynchronous activity of the neuronal network. To investigate the involvement of LGI1 in the regulation of the neuronal network, a knock-out mouse model *lgi1*$^{-/-}$ has been

DQ20170336751. This work has been developed within the BETPSY project, which is supported by a public grant overseen by the ANR (Agence nationale de la recherche), as part of the second "Investissements d'Avenir" program (reference ANR-18-RHUS-0012). EF was supported by a research grant from the Université Claude Bernard Lyon 1 attributed by the doctoral school Neurosciences et Cognition (ED476). The funders had no role in study design, data collection and analysis, decision to publish, or preparation of the manuscript.

**Competing interests:** The authors have declared that no competing interests exist.

**Abbreviations:** LGI1, leucine-rich glioma inactivated protein 1; ADLTE, autosomal dominant temporal lobe epilepsy; LE, limbic encephalitis; CNS, central nervous system; AMPA-R, α-amino-3-hydroxy-5-methyl-4-isoxazolepropionic acid receptors; IgG, Immunoglobulins G; Abs, antibodies; (a)CSF, (artificial) cerebrospinal fluid; ADAM, A disintegrin and metalloprotease protein; PSD, postsynaptic density; Kv, voltage-gated potassium channels; GAD2-RT, GAD2-rosa tomato mice; GAD, glutamic acid decarboxylase 65; ctrl, control; (e)EPSP, (evoked) excitatory postsynaptic potentials; LFP, local field potential; IO, input-output; STORM, stochastic optical reconstruction microscopy; 4-AP, 4-aminopyridine; DTX-K, dendrotoxin-K.

created [3]. In this model, severe epileptic seizures appeared from 2 weeks of age, leading to the death of the animal at three weeks post-natal, supporting the idea of an essential role of LGI1 during the maturation of the neuronal network [3]. Another study based on a transgenic mouse model carrying a genetic alteration associated with ADLTE, demonstrated that LGI1 has a crucial function on the maturation and pruning of excitatory synapses during the development [4]. Nevertheless, the role of LGI1 in the regulation of the mature neuronal network has been recently highlighted by the description of autoimmune encephalitis with LGI1 auto-antibodies (Abs) [5, 6]. LGI1-Abs have been found in the serum and the cerebrospinal fluid (CSF) of adult patients with limbic encephalitis (LGI1 LE) [6, 7] and such patient are also characterized by epileptic seizures. LGI1-Abs seem to play a direct role to block LGI1 protein indicating an essential function of LGI1 in the regulation of the neuronal network activity during adulthood [8–10].

Further studies investigating the function of LGI1 in the CNS found that LGI1 interacts at excitatory synapses with its transmembrane partners Disintegrin and metalloproteinase domain-containing protein 22 (ADAM22) and/or 23 (ADAM23) [11–14] to form a large trans-synaptic complex. By this complex, it was suggested that LGI1 regulates the expression and activity of voltage-gated potassium kv1.1 channels at the presynaptic compartment through its interaction with ADAM23 [3, 15–18], and the expression of α-amino-3-hydroxy-5-methyl-4-isoxazolepropionic acid receptors (AMPA-R) through its interaction with ADAM22 at the postsynaptic compartment [11]. Strangely, in contradiction with the epileptic seizures phenotype, the dysfunction of LGI1 was reported to reduce the expression of AMPA-R leading to the decrease of the excitatory synaptic transmission [3, 12, 16, 19]. Few regulatory mechanisms have been proposed so far, a study suggested that the main action of LGI1 would be on kv1.1 channels resulting in an homeostatic regulation [18], but this theory has not been proven yet while another theory suggested that the reduction of AMPA-R at excitatory synapses on inhibitory neurons would increase excitability and could explain the epileptic seizures [3]. Nevertheless, there is currently no demonstrated explanation for this contradiction.

The aim of our study was to investigate the synaptic function of LGI1 in the mature hippocampus. To do this, we used LGI1-Abs purified from the serum of patients with LGI1 LE and investigated the effects on AMPA-R expression on the regulation of the neuronal network activity. We showed that the neutralization of LGI1 protein by LGI1-Abs reduced AMPA-R expression at the surface of both excitatory and inhibitory neurons. Moreover, we observed that LGI1-Abs increased the hyperexcitability of the neuronal network independently of kv1.1 channels. Thus, we brought, for the first time, evidences that this increased hyperexcitability was due to a disturbance of the inhibitory network which is not able to control the overexcitability of the neuronal network.

## 2. Material and method

### Animals

The study was conducted in accordance with the European Community Council directive 2010/63/EU on the protection of animals used for experimental and scientific purposes. Animal care and treatment procedures were performed according to the ARRIVE guidelines approved by the French Ethical Committee of Lyon 1 University (#13703).

For the electrophysiological study, a total of 18 C57BL6/JRj mice aged of 7 weeks old (Janvier Labs) were used ($n_{Ctrl}$ = 6 mice, $n_{LGI1}$ = 7 mice, $n_{DTX-K}$ = 5 mice). Animals were placed at 12h/12h light/dark cycle with food and water *ad libitum*. For immunohistochemistry study, we used a transgenic GAD2-rosa tomato (GAD2-RT) mice model [20]. To do so,

Gad2-IRES-Cre$^{+/+}$ mice expressing Cre recombinase under the control of Gad2 promotor which encodes for the glutamic acid decarboxylase 65 (GAD65), were crossed with Gt(ROSA) 26Sor$^{tm9(CAG-tdTomato)Hze+/+}$ mice expressing the loxed stop codon into the *Gt(ROSA)26Sor* locus. Thus, 7 GAD2-RT mice expressing the fluorescent protein rosa tomato in GAD65 cells were used in this experiment.

## Immunoglobulins G purification

The serum of a patient with LGI1 LE was provided by the French Reference Center on Paraneoplastic Neurological Syndromes and Autoimmune Encephalitis and the NeuroBioTec biobank from Hospices Civils de Lyon. Control immunoglobulins G (ctrl IgG) were provided by the Etablissement Français du Sang (EFS).

Control and LE serum were incubated with protein-A coated beads (Protein A-Sepharose 4B Fast Flow, Sigma, #P9424) before being placed into chromatography columns (Evergreen Scientific, #208-3384-060). IgG were eluted with glycine (0.1M pH 2.8) and neutralized with Tris-HCL (1.5M, pH 8.8) before to be dialyzed (Dialysis Cassettes, Thermofisher, #87730) against PBS overnight at 4°C. Concentrations of purified IgG were determined by nanodrop assay.

## HEK239T cells immunostaining

HEK293T cells were seeded on glass coverslips (Labelians #LCO14), pretreated with 10 µg/µl poly-L-lysine (Sigma, #P1399). At 70% confluence, HEK293T cells were transfected with ADAM22 and LGI1-GFP plasmids using the lipofectamine LTK transfection kit (Thermofisher, #153388100). Transfected HEK293T cells were incubated with purified IgG for 1 hour at increasing dilution and then were fixed with 4% paraformaldehyde (PFA, Euromedex, #EM-15713-S). Cells were incubated for 30 minutes in a blocking buffer solution containing 1% Bovin Serum Albumine (BSA, Axday, #1000–70), 10% Normal Goat Serum (NGS, Eurobio, #S-100), PBS. Secondary antibodies against human IgG coupled with Alexa555 were incubated during 2 hours at room temperature. Cell cultures were stained with DAPI before to be mounted on slide with Fluoromount (Thermofisher, #4958–02). Images were acquired using an epifluorescence microscope (Zeiss, Imager Z1, Apotome).

## Immunocytofluorescence

Primary hippocampal cell cultures were incubated with purified ctrl or LGI1 IgG for 1 hour before to be fixed with 4% PFA (Euromedex, #EM-15713-S) for 15 minutes. After 3 washes in PBS, cells were incubated in blocking buffer solutions composed of 1% BSA (Axday, #1000–70), 10% NGS (Eurobio, #S-100), PBS for 1 hour. Secondary antibodies against human IgG coupled with Alexa555 (Invitrogen, #A21433) were incubated for 2 hours at room temperature. Cell nuclei were stained using 1 µg/ml DAPI (Sigma, #D9542) for 10 minutes before to be mounted in fluromount (Thermofisher, #4958–02). Images were acquired using epifluorescence microcope (Zeiss, Imager Z1 Apotome).

## Surgery and pumps implantation

6 weeks old mice were implanted with an osmotic pump (Alzet, 1007D) previously filled with purified ctrl or LGI1 IgG (60 µg/ml) or dendrotoxin-K (DTX-K) (100 nM). The day of surgery, mice were deeply anesthetized with isoflurane (IsoVet) 4% and the surgeries were performed on a stereotaxic frame. The cannula was implanted into the medial-septum according to the following coordinates: antero-posterior: Bregma+0.25 mm; lateral: Bregma+0.5mm; depth:

skull surface-3mm. Intra-peritoneal injection of metacam® (2mg/kg) were made immediately after the surgery and the next day. Mice brains were infused for 7 days before to proceed to the experiments.

## Electrophysiology

Acute hippocampal slices (400μm) were cut using a vibratome (HM 650V, ThermoFisher) in sucrose aCSF at 4°C composed of (mM): 250 Sucrose, 3 KCl, 1.25 $NaH_2PO_4$, 2 $CaCl_2$, 1 $MgCl_2$, 26 $NaHCO_3$, 10 D-Glucose saturated with 95% $O_2$ et 5% $CO_2$. Recordings were performed in regular aCSF of the following composition (in mM): 125 NaCl, 3 KCl, 1.25 $NaH_2PO_4$, 2 $CaCl_2$, 1 $MgCl_2$, 26 $NaHCO_3$, 10 D-Glucose saturated with 95% $O_2$ and 5% $CO_2$. On ipsilateral slices in regard to the cannula, we used 2–4 dorsal slices to perform experiments.

The evoked excitatory post-synaptic potentials (eEPSP) and the local field potentials (LFP) in the CA1 region of the hippocampus were recorded. For eEPSP recordings, a stimulating electrode (Phymep, #CBAR C75) was positioned on the Schaeffer collaterals. To establish the input-output (IO) curve, increasing stimulations (100 μA steps) between 0 to 600 μA for 0.05msec at 0.01Hz were delivered, and the slope of the evoked responses were analyzed. To study the global hyperexcitability of the network, LFP were recorded in the CA1 region of hippocampus. in presence of 4-aminopyridine (4-AP) (40μM, Tocris #0940) alone or 4-AP (40μM) and picrotoxin (100μM, Tocris #1128). The global hyperexcitability was determined by multiplying the numbers of ictus by the area of the ictus.

## PSD enrichment and immunoblot

Post-synaptic density (PSD) fractions were realized from hippocampal slices of infused mice with control purified IgG. Hippocampal slices were dissociated and centrifuged in buffer solution 1 composed of 0.32M sucrose, 10 mM HEPES, pH: 7.40 completed with protease inhibitor (Sigma, #4693132001) and orthovanadate (Biolabs, # P0758S). Crude membrane fractions in the pellets were collected and resuspended in the buffer solution 2 composed of 1mM EDTA, 4 mM HEPES, pH: 7.40, to chelate calcium, and centrifuged twice. The synaptosomal fractions in the pellet was resuspended and incubated in the buffer solution 3 composed of 100 mM NaCl, 20 mM HEPES, 0.5% Triton, pH 7.4 at 4°C. The solution was centrifuged and the non-PSD fraction contained in the supernatant was collected. The PSD fraction contained in the pellet was resuspended in the buffer solution 4 composed of 0.15 mM NaCl, 1% Triton, 1% DOC, 1%SDS, 20mM HEPES, pH: 7.50 and centrifuged to be collected.

Immunoblot were performed with the following antibodies: anti-ADAM22 (1:340 Abcam, # ab231340), anti-ADAM23 (1:5000 Abcam, # ab28304), anti-PSD-95 (1:1000 Cell Signaling, #3450S), anti-Synaptophysin (1:6000 Sigma, # S5768). Secondary antibodies conjugated to horseradish peroxidase (HRP) directed against mice (1:6000 JacksonLabs, #115-036-003) or rabbit (1:6000 JacksonLabs, #11-036-003) were used. Revelation and analysis have been realized with ImageLab software.

## Immunohistochemistry

GAD2-RT infused mice were anesthetized with isoflurane (IsoVet) and perfused with 5 ml NaCl 0.9% at 120 ml/hour then 5ml NaCl 0.9% at 60 ml/hour. The brain was removed and fixed into 4% PFA, (Euromedex, # EM-15713-S) for 24 hours and preserved in 30% sucrose solution for 72 hours. Then, brains were frozen into -40°C isopentane for 30 to 45 sec and embedded into optimal cutting temperature (OCT) compound. Coronal hippocampal sections (7μm) were cut with cryostat (Thermofisher, #NX50) at -20°C and were collected on high precision coverslips (Marienfield #0117580) previously coated with gelatin.

Coverslips were rehydrated in PBS during 5 min and incubated into blocking buffer 10% Normal Goat Serum (NGS, Eurobio, #S-100), 1% BSA (Axday, #1000–70). GluA1 rabbit antibodies (1:1000, Millipore #ABN241) were incubated overnight at 4˚C, following by rabbit Alexa647 antibodies (1:1000, Invitrogen #A21244).

## Super-resolution microscopy STORM

To perform super-resolution microscopy, stained slices fixed on coverslips were mounted in cavity slides (Marienfield # 631–9475) embedded into a switching buffer for dSTORM as reported [21, 22]. Image acquisitions were performed using a Nikon PALM-STORM microscope equipped with an EM-CCD camera (Andor Ixon 897 Ultra, gain 250). Acquisitions were made using the software NIS-Element. Analyzes were performed in the molecular layer of the dentate gyrus, and in the *stratum radiatum* and *stratum lacunosum molecular* of the CA3 and CA1 regions. Super-resolution acquisitions were performed using a 642 nm wavelength laser diode at 320 mW of the laser power and a Cy5 filter 4040C (M352119). GluA1 localization tabs were generated with ThunderStorm plugin on Image J software, and analyses were performed through SR Tesseler software.

## Analyze

Statistics were performed on GraphPad Prism 7 software. Data are represented with mean ±SEM. Normality conditions were assessed for every data with Shapiro-Wilk test. For the comparison of 3 groups, if data passed the normality, a one-way ANOVA was performed. If data did not pass the normality, nonparametric Kruskal-Wallis test was performed. For 2 groups comparison unpaired Student-t test were made if data passed the normality. If data did not pass the normality, Mann-Whitney test were performed. Data were considered statistically different when p-value $< 0.05$ ($^*$p$<0.05$; $^{**}$p$<0.01$; $^{***}$p$<0.001$).

# 3. Results

## 3.1. Purified IgG from patients with LGI1 LE contained LGI1 Abs

To evaluate LGI1 function in the adult hippocampus, we used purified IgG from the serum of a patient diagnosed with LGI1 LE. To ascertain if the purified IgG from the serum of patient with LGI1 LE contains LGI1 Abs, we performed an immunostaining on HEK293T cells expressing LGI1 protein with its partners ADAM22 to fix LGI1 at the surface of cells. We observed that purified IgG from LGI1 LE patient (LGI1 P1) stains the surface of cells expressing LGI1-GFP protein but not purified IgG from healthy subject (Control C1) (Fig 1). Thus, we confirmed that purified IgG from LGI1 LE patient recognized LGI1 protein on HEK293T cells and contained LGI1-Abs in opposition to IgG from healthy subject.

## 3.2. LGI1 Abs reduce the expression of AMPA receptors in the dentate gyrus of the hippocampus

To investigate if LGI1-Abs modulate the expression of AMPA-R at excitatory or inhibitory neurons, by STochastic Optical Reconstruction Microscopy (STORM) we analyzed the expression of GluA1 subunits of AMPA-R in hippocampal slices from mice infused with LGI1 or control IgG for 7 days. To distinguish excitatory and inhibitory neurons, we performed this study on hippocampal slices from mice expressing td-tomato in GAD65 interneurons.

First, we quantified the numbers of GluA1 subunits of AMPA-R inside cluster larger than 20 nm at the surface of inhibitory neurons expressing td-tomato (GAD+ cells) in the dentate gyrus (DG), CA3 and CA1 regions respectively (Fig 2A). In the DG, we observed that pre-

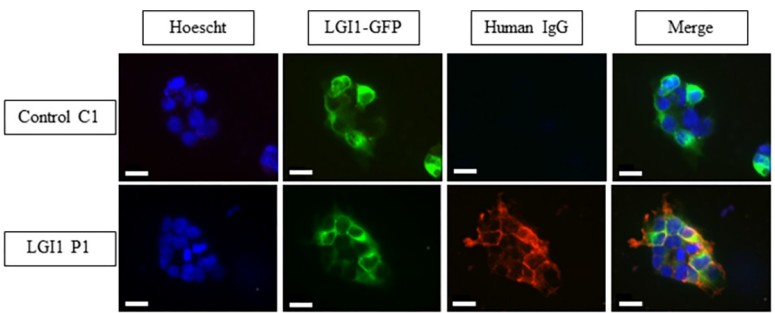

**Fig 1. Purified IgG from LGI1 LE patient contain LGI1 Abs.** HEK293T cells were transfected with LGI1-GFP and ADAM22 plasmids. Purified IgG from LGI1 LE patient are colocalized with LGI1-GFP signal at the cell surface of HEK293T transfected cells while no signal was found in cells treated with purified IgG from healthy subject. Scale bar = 20μm.

incubation with LGI1 IgG decrease significantly the relative surface number of GluA1 subunits compared to ctrl IgG. We did not observe any difference between LGI1 P1 and control C1 IgG infusion in the expression of GluA1 subunits in CA3 or CA1 regions (Fig 2B–2D). We observed that the density of GluA1 clusters was not different after LGI1 P1 or control C1 IgG pre-infusion in all the areas of the hippocampus considered (S2 Fig). These results indicate that pre-incubation with LGI1 P1 decreased the expression of AMPA-R at the surface of inhibitory neurons of the DG of the hippocampus but not in CA3 or CA1 area.

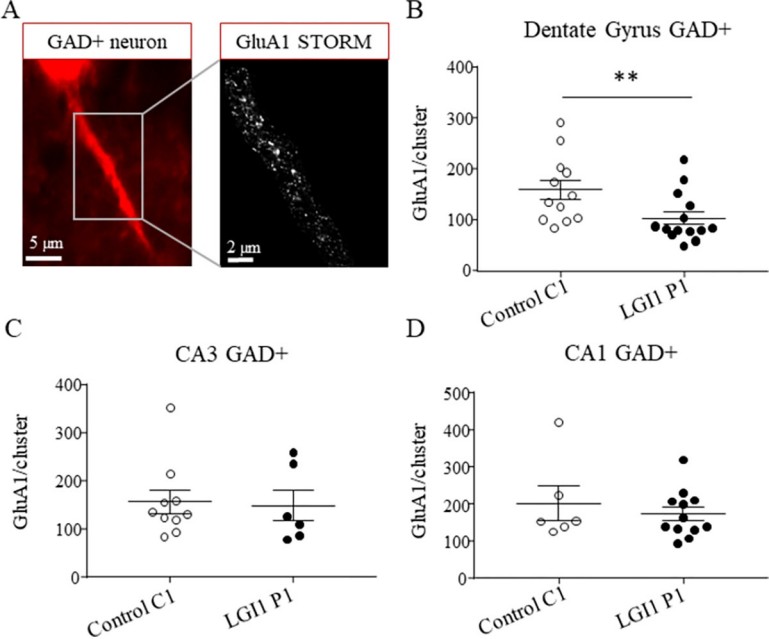

**Fig 2. LGI1 Abs decrease the expression of AMPA-R inside clusters at the surface of interneurons in the hippocampus.** Epifluorescence images (100x objective) of an inhibitory GAD+ neuron expressing td-tomato (left image). Magnification of the corresponding super-resolution STORM image showing the expression of GluA1 subunits along the neuronal extension (right image) (A). The total density of surface GluA1 subunits decreased after LGI1 P1 IgG infusion compared to control C1 IgG infusion at the surface of interneurons in the DG (Mann-Whitney **p = 0.0091, $n_{Ctrl}$ = 12 neurons, $n_{LGI1}$ = 14 neurons) (B) but not in CA3 region (Mann-Whitney test p = 0.63, $n_{Ctrl}$ = 11 neurons, $n_{LGI1}$ = 6 neurons) (C) nor CA1 region of the hippocampus (Mann-Whitney t-test p = 0.68, $n_{Ctrl}$ = 6 neurons, $n_{LGI1}$ = 12 neurons) (D). Data are represented as means ± SEM (analysis realized from $n_{Ctrl}$ = 4 mice, $n_{LGI1}$ = 3 mice).

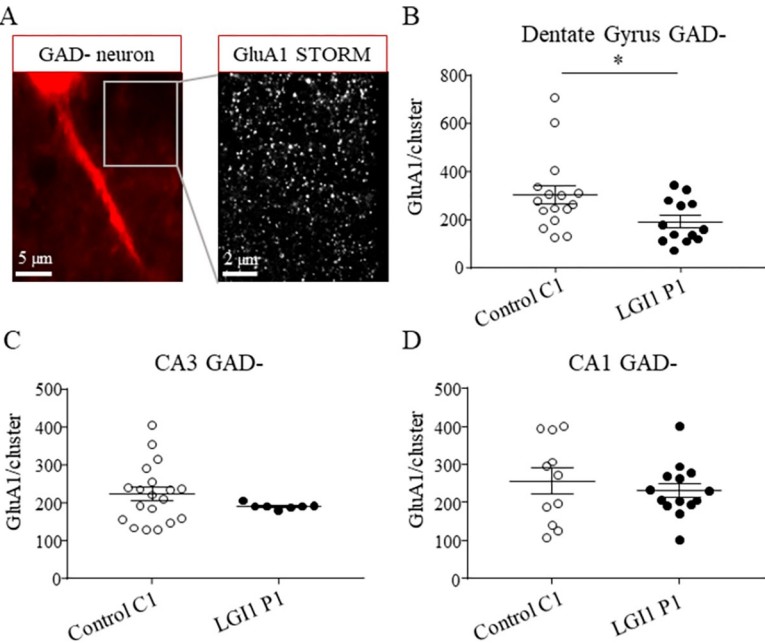

**Fig 3. LGI1 Abs decrease the expression of AMPA-R inside clusters at the surface of excitatory neurons in the hippocampus.** Epifluorescence images (100x objective) of GAD- region representing excitatory neurons (left image). The magnification showing the super-resolution STORM image of the expression of GluA1 subunits in the GAD- region (right image) (A). The total density of surface GluA1 subunits decreased after LGI1 P1 IgG infusion compared to control C1 IgG infusion in GAD- area in the DG of the hippocampus (Unpaired t test *p = 0.031, $n_{Ctrl =}$ 16 area, $n_{LGI1}$ = 13 area) (B) but not in CA3 region (Mann-Whitney test p = 0.33, $n_{Ctrl =}$ 19 area, $n_{LGI1}$ = 9 area) (C) nor CA1 (Unpaired t-test p = 0.49, $n_{Ctrl =}$ 11 area, $n_{LGI1}$ = 14 area) (D). Data are represented as mean ± SEM (analysis realized from $n_{Ctrl =}$ 4 mice, $n_{LGI1}$ = 3 mice.

Secondly, to investigate the expression of AMPA-R at the surface of excitatory neurons, we analyzed a GAD negative (GAD-) region in which GAD+ cells were absent (Fig 3A). We observed that LGI1 P1 IgG pre-incubation decreased the surface numbers of GluA1 subunits inside clusters in the DG but not in CA3 or CA1 area (Fig 3B–3D). We observed that there was no difference in the density of GluA1 clusters following LGI1 P1 or control C1 IgG pre-incubation in the DG, CA3 or CA1 area (S3 Fig). Altogether, our results indicate that LGI1-Abs decrease the expression of AMPA-R at the surface of inhibitory and excitatory neurons only in the dentate gyrus of the hippocampus.

### 3.3. LGI1 Abs increase the hyperexcitability of the neuronal network independently of kv1.1 channels inhibition

The blockade of LGI1 function leads to epileptic seizures [3]. To investigate the effect of LGI1 IgG on the hippocampal neuronal network transmission, we recorded by electrophysiology, the local field potentials (LFP) in CA1 area at the exit of the hippocampal network of acute hippocampal slices mice infused during 7 days with control or LGI1 IgG. First, to determine if the basal transmission was modulated by LGI1 IgG pre-infusion compared to control IgG, we recorded the evoked excitatory post-synaptic potentials (eEPSP) in CA1 area when delivering increasing voltage pulses at Schaffer collaterals. The resulting input-output (IO) curve obtained by analyzing the slope of eEPSP, was not different between LGI1 or control IgG pre-infusion (Fig 4A). This result means that there is no effect of LGI1 IgG pre-infusion on the basal transmission compared to control IgG. Then, we recorded the spontaneous local field

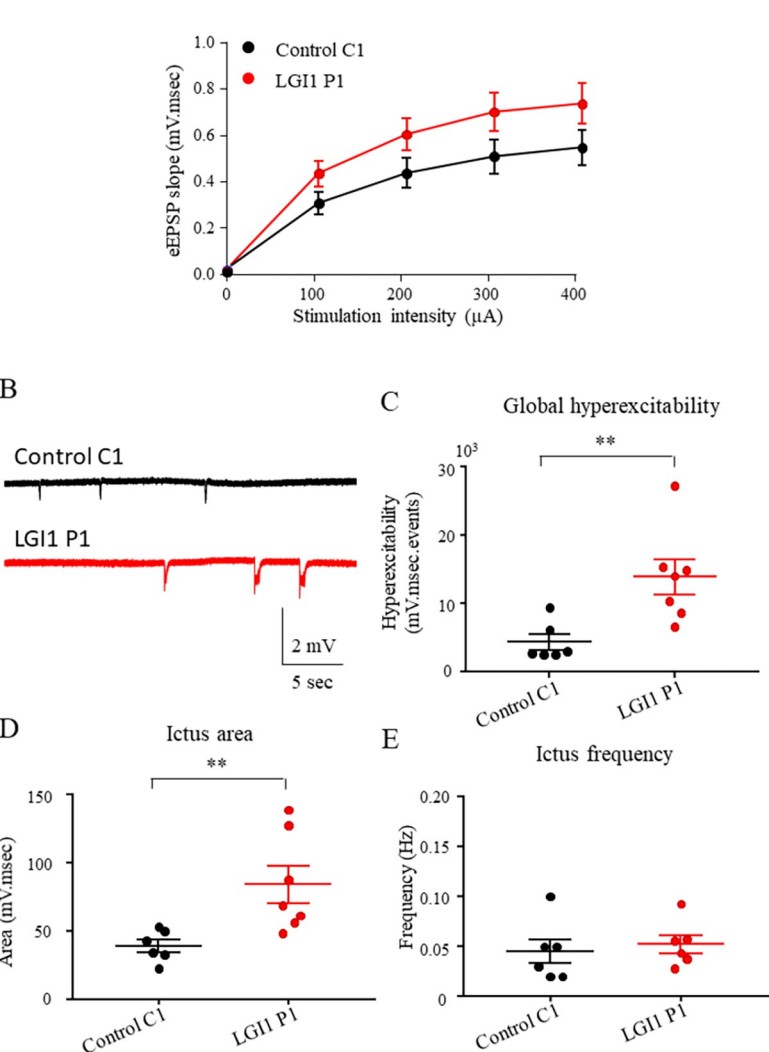

**Fig 4. LGI1 abs increase induced neuronal hyperexcitability in the hippocampus.** LFP were recorded in CA1 region of acute hippocampal slices from mice infused for 7 days with control C1 IgG, LGI1 P1 IgG. Increasing stimulations intensity were delivered to Schaffer collaterals and the slope of eEPSP in CA1 area were plotted. No effect of the infusion of control C1 IgG, LGI1 P1 IgG on the IO curve was detected (two-way ANOVA test, p = 0.27; $n_{Ctrl}$ = 12; $n_{LGI1}$ = 14) (A). Examples of ictus recorded in slices infused with control C1 IgG, LGI1 P1 IgG (B). The global hyperexcitability of the neuronal network is increased after LGI1 P1 IgG infusion compared to control C1 IgG infusion (Kruskal-Wallis test *p = 0.02, Dunn's *post Hoc* test LGI1-Ctrl **p = 0.0082) (C) as well as the area of the ictus (Kruskal-Wallis test *p = 0.02, Dunn's *post Hoc* test LGI1-Ctrl **p = 0.12) (D) while frequency is unchanged (Kruskal-Wallis test *p = 0.40, Dunn's *post Hoc* test LGI1-Ctrl **p>0.99) (E). Data are represented as mean ± SEM ($n_{Ctrl}$ = 6 mice, $n_{LGI1}$ = 7 mice).

potential in the CA1 area of the hippocampus in order to consider the output of the hippocampal network. In basal condition, we did not observe any spontaneous epileptic activity in hippocampal slices from control or LGI1 infused mice. Thereby, we added 4-aminopyridine (4-AP) (40μM), a non-specific blocker of voltage gated potassium channels to induce an hyperexcitability of the hippocampal neuronal network. The addition of 4-AP induced an epileptic like activity consisting in spontaneous depolarizations similar to ictus (Fig 4B). We observed that the induced hyperexcitability was significantly increased in slices from mice pre-

infused with LGI1 IgG when compared to slices pre-infused with control IgG (Fig 4C). More specifically, we observed that LGI1 IgG increased the ictus area (Fig 4D) but not the ictus frequency (Fig 4E) when compared to control IgG. These results indicate that LGI1-Abs increased the hyperexcitability of the neuronal network mainly by increasing the cell synchronization compared to the ctrl IgG. Similar results were obtained with purified IgG from a second patient (S1 and S2 Figs).

We investigated if a chronic inhibition of kv1.1 channels would trigger the same alterations on the neuronal network than the LGI1 P1 IgG pre-infusion. To do this, we infused mice with dendrotoxin-K (DTX-K) (100 nM) for 7 days and record the LFP in CA1 region of acute hippocampal slices from these mice. We recorded eEPSP in the CA1 area of hippocampus after increasing stimulation of Schaffer collaterals and we did not observe any effect of DTX-K infusion on the IO curve compared to LGI1 P1 nor control C1 IgG infusion (Fig 5A). We observed that the global hyperexcitability after DTX-K infusion was not different compared to control C1 IgG, when 4-AP was added to create an hyperexcitability of the hippocampal neuronal network (Fig 5B and 5C). When looking at area and frequency of the ictus we observed that the ictus area was not different following DTX-K infusion compared to control C1 IgG infusion (Fig 4D) as well as the frequency of ictus (Fig 5D) although a tendency of increase could be seen. All together these results indicate that the blockage of kv1.1 channels by DTX-K infusion in the hippocampus of mice does not increase the global hyperexcitability.

### 3.4. LGI1 Abs reduced the inhibitory network transmission

We showed that LGI1-Abs increased the induced hyperexcitability of the hippocampal neuronal network, independently of kv1.1 channels down-regulation. To investigate if the increased hyperexcitability of LGI1 IgG could result from a down-regulation of the inhibitory network, we recorded the LFP in the CA1 area of acute hippocampal slices from mice pre-infused with LGI1 P1 or control C1 IgG in presence of 4-AP (40 μM) and picrotoxin (100 μM), a GABA$_A$ antagonist. In such conditions we did not observe any difference in the global hyperexcitability of the neuronal network between LGI1 P1 and control C1 IgG pre-infusion (Fig 6). More specifically, the blockade of the inhibitory network increases the global hyperexcitability of the neuronal network infused for control C1 IgG compared to the non-blockade of the inhibitory network, while the blockade of the inhibitory network did not change the global hyperexcitability of the neuronal network infused with LGI1 P1 IgG compared to the non-blockade of the inhibitory network. Similar results were obtained with purified IgG of a second patient (S3 Fig). These results indicate that the LGI1-Abs impaired the inhibitory network in the hippocampus which results in the increase of the hyperexcitability of the neuronal network.

## 4. Discussion

Previous studies reported a decrease of AMPA-R expression when LGI1 function is altered [3, 12, 16, 19]. Our work confirmed that LGI1-Abs reduced the expression of AMPA-R and additionally, we showed that this reduction of expression affects both excitatory and inhibitory neurons of the DG and no other area of the hippocampus. Moreover, we found that LGI1-Abs increased the hyperexcitability of the neuronal network certainly due to a down-regulation of the inhibitory neuronal network. Altogether, our results suggest that the reduction of AMPA-R triggers a more drastic downregulation of the inhibitory transmission compared to the excitatory transmission which creates an imbalance between excitatory and inhibitory networks. This imbalance prevents the inhibitory network to control hyperexcitability of the neuronal network. Our study suggests that the inhibitory network down-regulation is a mechanism probably involved in the generation of epileptic seizures in patients with LGI1 LE.

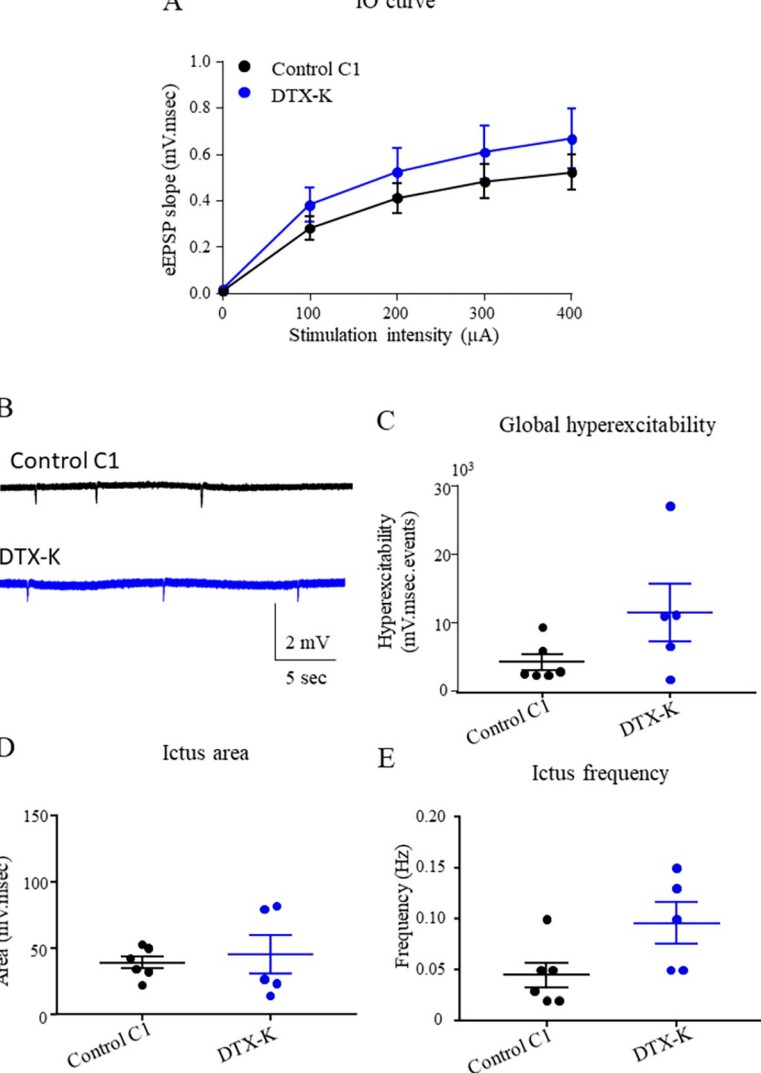

**Fig 5. DTX-K infusion does not increase neuronal hyperexcitability in the hippocampus.** LFP were recorded in CA1 region of acute hippocampal slices from mice infused for 7 days with control C1 IgG or DTX-K. Increasing stimulations intensity were delivered to Schaffer collaterals and the slope of eEPSP in CA1 area were plotted. No effect of the infusion of control C1 IgG vs DTX-K on the IO curve was detected (two-way ANOVA test, p = 065; $n_{Ctrl}$ = 12; $n_{DTX-K}$ = 8) (A). Examples of ictus recorded in slices infused with control C1 IgG or DTX-K (B). The global hyperexcitability of the neuronal network is not increased significantly after DTX-K infusion compared to control C1 IgG infusion (Dunn's *post Hoc* test DTX-K- Ctrl p = 0.11) (C) as well as the area of the ictus (Fisher LSD *post Hoc* test, Ctrl-DTX-K p = 0.73) (D) while frequency is not significantly increased (Kruskal-Wallis test, p = 0.09) (E). Data are represented as mean ± SEM ($n_{Ctrl}$ = 6 mice, $n_{DTX-K}$ = 5 mice).

The reduction of AMPA-R in the DG of the hippocampus after LGI1 IgG infusion is in accordance with the higher expression profile of LGI1 in the DG area compared to CA3 or CA1 area of the hippocampus [2, 3, 23]. The effect of an infusion of LGI1 IgG on the expression of AMPA-R in the hippocampus has been investigated in a previous study in which it was reported a reduction in the DG and CA1 regions after 18 days of Abs infusion [16]. Nevertheless, in this study, the infusion was performed into the ventricles and acquisitions were realized with confocal microscopy. In our study, we infused IgG directly into the septum, a structure nearby the hippocampus, and we quantified the AMPA-R expression with a better resolution

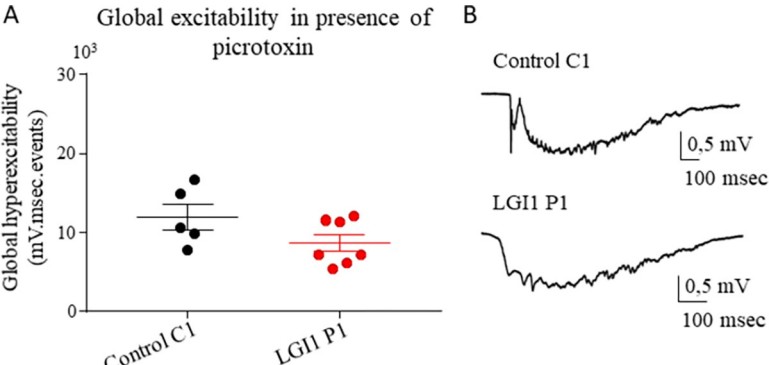

**Fig 6. LGI1 abs have no effect on the global hyperexcitability when the inhibitory network was blocked.** (A) LFP in presence of 4-AP (40µM) and picrotoxin (100µM) were recorded in CA1 area of acute hippocampal slices from mice infused for 7 days with control C1 or LGI1 P1 IgG. No difference was found in the global hyperexcitability between control C1 IgG and LGI1 P1 IgG infused neuronal network when recordings were performed in presence of picrotoxine (Mann-Whitney test p = 0.20, $n_{Ctrl}$ = 6 mice, $n_{LGI1}$ = 7 mice). (B) Examples of ictus recorded in presence of 4-AP and picrotoxine. Data are represented as mean ± SEM ($n_{Ctrl}$ = 6 mice, $n_{LGI1}$ = 7 mice).

using STORM microscopy. The goal of our study was to understand how a reduction of AMPA-R expression leads to epileptic seizures. In our electrophysiological experiments, we blocked the potassium channels using 4-AP to create hyperexcitability of the neuronal network and to prevent a functional effect caused by a potential kv1.1 channels alteration. We observed that LGI1 IgG increase the hyperexcitability of the neuronal network when kv channels were blocked. Nevertheless, some authors suggested that the alteration on AMPA-R could be due to a homeostatic regulation of kv1.1 channels [18]. Our study does not support this hypothesis. Indeed, although the blocking of Kv1.1 channels by 4-AP at the concentration we used is likely incomplete, we highlighted that a prolonged kv1.1 channels blockage by DTX-K infusion, did not result in an increased neuronal hyperexcitability when kv channels were blocked by 4-AP. Although we cannot rule out an action of LGI1 IgG on Kv1.1 channels, the results obtained in our specific model underline a specific role of LGI1 in the regulation of AMPA-R independently of kv1.1 channels reported reduction [16, 18].

We also demonstrated a decrease of AMPA-R expression on inhibitory and excitatory neurons of the DG. It is important to note that we analyzed the total surface expression of AMPA-R without focusing on the synaptic structure. Indeed, lack of a good combination of primary and secondary antibodies prevented us to investigate synaptic AMPA-R in the hippocampus by super-resolution microscopy in hippocampal slices of infused mice. A recent study showed that the synaptic interactions between LGI1 and ADAM22 are essential for the alignment of presynaptic and postsynaptic compartment involving a large complex of synaptic proteins, including AMPA-R [24]. Thereby, it will be interesting to investigate synaptic AMPA-R organization in the hippocampus of mouse infused with LGI1 IgG.

Patients with LGI1 LE exhibited epileptic seizures known to be due to an hyperexcitability and hypersynchronous activity of the neuronal network [8–10]. Inhibitory network disturbance has already been shown to drive neuronal network hyperexcitability especially in the DG [25–27]. Indeed, interneurons are essential for the control of the balance of neuronal transmission [28, 29]. In our study, we suggest a role of the inhibitory network in the epileptic seizures observed in patient with LGI1 LE. Even if the expression of AMPA-R is reduced at the surface of both excitatory and inhibitory neurons of the hippocampus, the high arborization of interneurons allow them to contact many excitatory neurons. For example, it was shown that one basket cell can contact 2,500 pyramidal cells and that hilar perforant path associated cell

(HIPP cells) may generate 100,000 synapses on the DG [30, 31]. Moreover, the dentate granule cells project mostly on inhibitory neurons compared to excitatory neurons [32]. Thus, a reduction of excitatory inputs on CA3 region provide an increase of the overall excitability due to an imbalance between excitation and inhibition transmission. Thus, these arguments can explain why small alterations of interneurons functioning may lead to high consequences of the neuronal network balance. Different subtypes of interneurons composed the hippocampal network [31] and the role of LGI1 in these different sub-populations remain to be elucidated. Further studies are needed to clarify the exact mechanisms underlying the dysregulation of the inhibitory network by LGI1 abs.

## 5. Conclusion

To conclude, we demonstrated an essential role of LGI1 in the regulation of AMPA-R expression and revealed a functional role of LGI-1 mainly on the inhibitory network. More studies are needed to clarify the underlying mechanisms of the dysregulation observed in LGI1 LE patients but our results indicate that special attention should be given to the inhibition as a leading cause for epilepsy in these patients.

## Supporting information

**S1 Fig. Purified IgG from LGI1 LE patient 2 5P2) contain LGI1 Abs.** HEK293T cells were transfected with LGI1-GFP and ADAM22 plasmids. Purified IgG from LGI1 LE patient 2 are colocalized with LGI1-GFP signal at the cell surface of HEK293T transfected cells while no signal was found in cells treated with purified IgG from healthy subject. Scale bar = 20μm. (TIF)

**S2 Fig. LGI1 abs from P2 increase induced neuronal hyperexcitability in the hippocampus.** LFP were recorded in CA1 region of acute hippocampal slices from mice infused for 7 days with control C1 IgG, LGI1 P2. (A) Increasing stimulations intensity were delivered to Schaffer collaterals and the slope of eEPSP in CA1 area were plotted. No effect of the infusion of control C2 IgG, LGI1 P2 on the IO curve was detected (two-way ANOVA test, p = 0,27; $n_{Ctrl}$ = 12; $n_{LGI1}$ = 14). (B) Examples of ictus recorded in slices infused with control C2 IgG, LGI1 P2 IgG. (C) The global hyperexcitability of the neuronal network was significantly increased on slices infused with LGI1 P2 IgG compared to slices infused with control C2 IgG (** for p = 0,0023, Mann Whitney test, nC2 = 9 souris nP2 = 11 mice). (D) The ictus area was significantly increased between slices infused with LGI1 P2 IgG and slices infused with control C2 IgG (*** for p = 0,0004; nC2 = 9 nP2 = 11 mice). (E) The ictus frequency was not different between LGI1 P2 IgG, control C2 IgG (Mann Whitney test: p = 0,44; nC1 = 9, nP1 = 11 mice). Data are represented as mean ± SEM. (TIF)

**S3 Fig. LGI1 abs have no effect on the global hyperexcitability when the inhibitory network was block.** (A) LFP in presence of 4-AP (40μM) and picrotoxin (100μM) were recorded in CA1 area of acute hippocampal slices from mice infused for 7 days with control C2 or LGI1 P2 IgG. No difference was found in the global hyperexcitability between control C2 IgG and LGI1 P2 IgG infused neuronal network when recordings were performed in presence of picrotoxine (Mann-Whitney test, p = 0,80; nC2 = 8; nP2 = 8 mice). Data are represented as mean ± SEM. (TIF)

**S1 Data.** (XLSX)

**S2 Data.**
(XLSX)

**S3 Data.**
(XLSX)

## Acknowledgments

We thank Camilla Luccardini, Bruno Chapuis and Denis Ressnikoff from the Centre d'Imagerie Quantitative Lyon-Est (CIQLE), Alexandra Vandermoeten from the Animalerie SPF SCAR at the Faculté de Médecine et de Pharmacie Rockefeller. We thank Magali Mondin from Bordeaux Imaging Center for helping in the development of an analysis program on SR Tesseler software.

## Author Contributions

**Conceptualization:** Olivier Pascual.

**Data curation:** Elodie Fels.

**Formal analysis:** Elodie Fels.

**Funding acquisition:** Jérôme Honnorat.

**Investigation:** Elodie Fels, Marie-Eve Mayeur, Estelle Wayere, Clémentine Vincent, Céline Malleval.

**Methodology:** Elodie Fels, Céline Malleval, Olivier Pascual.

**Resources:** Marie-Eve Mayeur, Estelle Wayere, Clémentine Vincent, Céline Malleval.

**Supervision:** Jérôme Honnorat, Olivier Pascual.

**Validation:** Jérôme Honnorat, Olivier Pascual.

**Writing – original draft:** Elodie Fels.

**Writing – review & editing:** Jérôme Honnorat, Olivier Pascual.

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
