## [Decision Letter · Decision Letter 0]

22 Apr 2022

PONE-D-22-03323Dysregulation of the hippocampal neuronal network by LGI1 auto-antibodies.PLOS ONE

Dear Dr. Pascual,

Thank you for submitting your manuscript to PLOS ONE. After careful consideration, we feel that it has merit but does not fully meet PLOS ONE’s publication criteria as it currently stands. Therefore, we invite you to submit a revised version of the manuscript that addresses the points raised during the review process.

We look forward to receiving your revised manuscript.

Kind regards,

Giuseppe Biagini, MD

Academic Editor

PLOS ONE

Journal Requirements:

“This work was supported by FRM (Fondation pour la recherche médicale) DQ20170336751. This work has been developed within the BETPSY project, which is supported by a public grant overseen by the ANR (Agence nationale de la recherche), as part of the second “Investissements d’Avenir” program (reference ANR-18-RHUS-0012). Elodie Fels was supported by a research grant from the Université Claude Bernard Lyon 1 attributed by the doctoral school Neurosciences et Cognition (ED476).”

We note that you have provided additional information within the Funding Section that is not currently declared in your Funding Statement. Please note that funding information should not appear in the Funding section or other areas of your manuscript. We will only publish funding information present in the Funding Statement section of the online submission form.

“This work was supported by FRM (Fondation pour la recherche médicale) DQ20170336751. This work has been developed within the BETPSY project, which is supported by a public grant overseen by the ANR (Agence nationale de la recherche), as part of the second “Investissements d’Avenir” program (reference ANR-18-RHUS-0012). Elodie Fels was supported by a research grant from the Université Claude Bernard Lyon 1 attributed by the doctoral school Neurosciences et Cognition (ED476).”

“This work was supported by FRM (Fondation pour la recherche médicale) DQ20170336751. This work has been developed within the BETPSY project, which is supported by a public grant overseen by the ANR (Agence nationale de la recherche), as part of the second “Investissements d’Avenir” program (reference ANR-18-RHUS-0012). Elodie Fels was supported by a research grant from the Université Claude Bernard Lyon 1 attributed by the doctoral school Neurosciences et Cognition (ED476).”

Please include this amended Role of Funder statement in your cover letter; we will change the online submission form on your behalf

Reviewers' comments:

Reviewer's Responses to Questions

**Comments to the Author**

1. Is the manuscript technically sound, and do the data support the conclusions?

Reviewer #1: Yes

Reviewer #2: Yes

2. Has the statistical analysis been performed appropriately and rigorously? 

Reviewer #1: I Don't Know

Reviewer #2: Yes

3. Have the authors made all data underlying the findings in their manuscript fully available?

Reviewer #1: Yes

Reviewer #2: No

4. Is the manuscript presented in an intelligible fashion and written in standard English?

Reviewer #1: Yes

Reviewer #2: Yes

5. Review Comments to the Author

Reviewer #1: This Ms is quite clear with sufficient data in introduction to understand the project.

At first glance, the results sound good. However this is not always easy to find the n numbers for each experiment. Are you sure that the concentration of 4AP is sufficient to fully blocks the K channels? At least you should discuss this point.

I was particularly dispapointed by the methods section that lacks a lot of sufficient precisions.

I have made a short list to give some few examples:

1) Ln98 “animals” section: please give the n numbers for each mouse category.

2) Ln 160 “Surgery and pumps implantation”. Please give more details about the isoflurane rate.

3) For the IO curves, what was the step when you increased the intensity of stimulation?

4) In a lot of places, the concentrations (in particular for the antibodies) are not indicated.

5) Please be consistent and precise: sometimes you use “.” and sometimes “,” as a decimal separator. The both can have completely different meanings and are not equivalent.

Several words are written in French but not in English.

For example to name a few:

1) Immunoglobulines or Synaptophysine (ln200)

2) blockage (Ln325).

Some other sentences are not correct:

1) Ln117: should be “before being placed”.

2) Or Ln177 “a stimulating electrode (Phymep, #CBAR C75) was position on the Schaeffer collaterals” please correct.

Other minor comment: Ln73-74: the full name of ADAM22 is “Disintegrin and metalloproteinase domain-containing protein 22”

Reviewer #2: It was an interesting study, and the authors collected precious data, which I think, with some work, will be a valuable contribution to the literature to be published. The manuscript is easy and understandable to follow, and the authors obtained enough data to support the conclusion. There are some issues as follows:

2. Inflammatory enzymes have been observed in the brains of people with epilepsy. What do you think about the contribution of inflammation to hyperexcitability?

3. As the cannula was implanted into the medial-septum, purified ctrl, LGI1 IgG, or dendrotoxin-K were spread in all parts of the hippocampus. Please mention the total number of slices used in this study and which parts of the hippocampus were selected for experiments.

4. LGI1 is necessary for correctly positioning proteins (such as ADAM22 and ADAM23) at the synapse. Do you think in LGI1 (-/-) the concentration of these proteins will be changed?

5. Do you think acute lockdown of the LGI1 affects the cellular viability and number of active excitatory synapses in primary cultures?

6. Why did the authors select Wistar rats for the primary hippocampal cell culture section?

6. Supplementary data S4 and S5 are mentioned in the manuscript, but they are missing.

7. Recent studies begin to reveal a trans-synaptic configuration of the LGI1-ADAM22 complex and its pivotal role in AMPA and NMDA receptor-mediated synaptic transmission through regulating MAGUKs. Do you think NMDA receptors could affect the results?

6. PLOS authors have the option to publish the peer review history of their article (what does this mean?). If published, this will include your full peer review and any attached files.

Reviewer #1: No

Reviewer #2: No

---

## [Author Response · Author response to Decision Letter 0]

31 May 2022

response to the reviewers:

Reviewer #1: This Ms is quite clear with sufficient data in introduction to understand the project.

At first glance, the results sound good. However this is not always easy to find the n numbers for each experiment. Are you sure that the concentration of 4AP is sufficient to fully blocks the K channels? At least you should discuss this point. 

We thank the reviewer to raise this important concern, indeed the concentration we used do not totally block Kv1.1 and kv1.2 channels, this is the reason why we also performed infusion of DTX-k which is more specific and that we used at high concentration. This point is now discussed in the discussion as follow. “Indeed, although the blocking of KV1.1 channels by 4-AP at the concentration we used is likely incomplete, …”

I was particularly dispapointed by the methods section that lacks a lot of sufficient precisions. I have made a short list to give some few examples: 1) Ln98 “animals” section: please give the n numbers for each mouse category.

We have now added the n for each category.

2) Ln 160 “Surgery and pumps implantation”. Please give more details about the isoflurane rate.

The isoflurane rate is now added in the material and methods section.

3) For the IO curves, what was the step when you increased the intensity of stimulation?

The step was 100µA, it is now added in the material and methods

4) In a lot of places, the concentrations (in particular for the antibodies) are not indicated.

The concentration of antibodies used as been added to the text in the material and method section.

5) Please be consistent and precise: sometimes you use “.” and sometimes “,” as a decimal separator. The both can have completely different meanings and are not equivalent.

This has been corrected

Several words are written in French but not in English. For example to name a few: 

1) Immunoglobulines or Synaptophysine (ln200)

2) blockage (Ln325)

Typo errors have been corrected. Thanks a lot for the thorough full reading.

Some other sentences are not correct:

1) Ln117: should be “before being placed”.

This has been corrected 

2) Or Ln177 “a stimulating electrode (Phymep, #CBAR C75) was position on the Schaeffer collaterals” please correct.

This has been corrected

Other minor comment: Ln73-74: the full name of ADAM22 is “Disintegrin and metalloproteinase domain-containing protein 22”

It has been corrected. We would like to thank the reviewer for his/her reading and the corrections he/she provided.

Reviewer #2: It was an interesting study, and the authors collected precious data, which I think, with some work, will be a valuable contribution to the literature to be published. The manuscript is easy and understandable to follow, and the authors obtained enough data to support the conclusion. There are some issues as follows:

2. Inflammatory enzymes have been observed in the brains of people with epilepsy. What do you think about the contribution of inflammation to hyperexcitability? 

In our case, although we have not measured carefully the inflammation in our model we could not see obvious signs of inflammation such as change of microglial and astrocytes shapes. It should be mentioned at this point that we could not detect any spontaneous seizures in the mice and could not detect obvious neuronal loss although not measured in vivo. We believe that the infusion of autoantibodies, only induced a mild change of network excitability compared to status epilepticus and may thus not trigger inflammation. 

3. As the cannula was implanted into the medial-septum, purified ctrl, LGI1 IgG, or dendrotoxin-K were spread in all parts of the hippocampus. Please mention the total number of slices used in this study and which parts of the hippocampus were selected for experiments.

Thank you for raising this point. Although we found a very homogeneous diffusion of the antibodies in both hemispheres, we used only ipsilateral slices and mostly the dorsal part of the hippocampus. We have now added those precisions to the material and methods section as follow: “On ipsilateral slices in regard to the cannula, we used 2-4 dorsal slices to perform experiments”

4. LGI1 is necessary for correctly positioning proteins (such as ADAM22 and ADAM23) at the synapse. Do you think in LGI1 (-/-) the concentration of these proteins will be changed?

This is a very interesting question indeed. Fukata and colleagues in 2010, observed a reduced expression of ADAM22 and ADAM23 proteins at the membrane of cells in the hippocampus from transgenic mice lgi1 -/- compared to wild type mice. However, this has never been studied in a model of mice infused with anti-LGI1 IgGs. In respect to the results we presented in this work, we could expect the surface expression of ADAMs would remain stable for a while but their locations may change. This misallocation could finally end up by an internalization fitting with the observation made by Fukata and colleagues of a global decrease of concentration of ADAMs.

5. Do you think acute lockdown of the LGI1 affects the cellular viability and number of active excitatory synapses in primary cultures?

We have performed some experiments on hippocampal primary cell cultures to evaluate cell viability in presence of autoantibodies. We performed LDH assays to quantify neuronal death but we could not detect significant neuronal loss following incubation anti LGI1 IgG for 48 hours when compared to controls. We also performed immunostaining and electrophysiology and could not found any obvious alteration of synapses or synapses number. We did not go further into the analysis of primary cultures as we found that they were very heterogeneous in term of activity. 

6. Why did the authors select Wistar rats for the primary hippocampal cell culture section?

This section should not appear in the manuscript, as we did not provide results from cell cultures due to their heterogeneity we were mentioning earlier. we apologies for the confusion and it is now removed from the material and method section. To be transparent on the matter, we initially started experiments on mice cultures but we had very heterogeneous results in electrophysiology. Because we suspected the quality of our culture to be responsible for these variable results we switched to rat cultures that were supposed to be more robust. This switch from mice to rat did not improve the heterogeneity of the neuronal activity found in our cultures. We thus decided to discard those results and focused on the results we obtained on slices.

6. Supplementary data S4 and S5 are mentioned in the manuscript, but they are missing.

We are sorry about this issue they were mislabeled and were supposed to refer to S2 and S3. This point has been corrected in the new version of the manuscript

7. Recent studies begin to reveal a trans-synaptic configuration of the LGI1-ADAM22 complex and its pivotal role in AMPA and NMDA receptor-mediated synaptic transmission through regulating MAGUKs. Do you think NMDA receptors could affect the results?

This is a very interesting question. We did not quantify the modification of NMDA receptors in our experiments but it is not excluded that NMDA receptors are altered too. Our conclusion did not confer a unique role of LGI1 on AMPA receptors but allow us to assess that LGI1 blockade leads to the reduction of AMPA receptors. It is not excluded that other proteins of the complex would be altered too. Indeed, Fukata and colleagues in 2021 showed that there are many proteins involved in the LGI 1-ADAM22 complex included NMDA-receptors. Thus, we cannot rule out that LGI1 antibodies, also affect NMDA receptors. In future studies, the question should address which proteins are affected by LGI1 blockage and how this dysregulation occurs. 

---

## [Decision Letter · Decision Letter 1]

18 Jul 2022

Dysregulation of the hippocampal neuronal network by LGI1 auto-antibodies.

PONE-D-22-03323R1

Dear Dr. Pascual,

We’re pleased to inform you that your manuscript has been judged scientifically suitable for publication and will be formally accepted for publication once it meets all outstanding technical requirements.

Kind regards,

Giuseppe Biagini, MD

Academic Editor

PLOS ONE

Additional Editor Comments (optional):

Reviewers' comments:

Reviewer's Responses to Questions

**Comments to the Author**

1. If the authors have adequately addressed your comments raised in a previous round of review and you feel that this manuscript is now acceptable for publication, you may indicate that here to bypass the “Comments to the Author” section, enter your conflict of interest statement in the “Confidential to Editor” section, and submit your "Accept" recommendation.

Reviewer #2: All comments have been addressed

Reviewer #3: All comments have been addressed

2. Is the manuscript technically sound, and do the data support the conclusions?

Reviewer #2: Yes

Reviewer #3: Yes

3. Has the statistical analysis been performed appropriately and rigorously? 

Reviewer #2: Yes

Reviewer #3: Yes

4. Have the authors made all data underlying the findings in their manuscript fully available?

Reviewer #2: Yes

Reviewer #3: Yes

5. Is the manuscript presented in an intelligible fashion and written in standard English?

Reviewer #2: Yes

Reviewer #3: Yes

6. Review Comments to the Author

Reviewer #2: The authors answered my comments clearly and also included or removed the parts that needed to be modified.

Reviewer #3: After adding the changes requested by the first two reviewers, the article seems suitable for publication.

7. PLOS authors have the option to publish the peer review history of their article (what does this mean?). If published, this will include your full peer review and any attached files.

Reviewer #2: No

Reviewer #3: No

---

## [Editor Report · Acceptance letter]

11 Aug 2022

PONE-D-22-03323R1 

Dysregulation of the hippocampal neuronal network by LGI1 auto-antibodies. 

Dear Dr. Pascual:

I'm pleased to inform you that your manuscript has been deemed suitable for publication in PLOS ONE. Congratulations! Your manuscript is now with our production department. 

Kind regards, 

on behalf of

Dr. Giuseppe Biagini 

Academic Editor

PLOS ONE